# Survival Benefits of Chemotherapy for Patients with Advanced Pancreatic Cancer in A Clinical Real-World Cohort

**DOI:** 10.3390/cancers11091326

**Published:** 2019-09-07

**Authors:** Maximilian Kordes, Jingru Yu, Oscar Malgerud, Maria Gustafsson Liljefors, J. -Matthias Löhr

**Affiliations:** 1Department of Clinical Science, Intervention and Technology, Karolinska Institute, 171 77 Stockholm, Sweden (M.K.) (O.M.); 2Upper Gastrointestinal Unit, Cancer Division, Karolinska University Hospital, 171 76 Stockholm, Sweden; 3Department of Medical Epidemiology and Biostatistics, Karolinska Institute, 171 77 Stockholm, Sweden; 4Department of Oncology-Pathology, Karolinska Institute, 171 76 Stockholm, Sweden

**Keywords:** pancreatic adenocarcinoma, gemcitabine, nab-paclitaxel, capecitabine, FOLFIRINOX, real-world data, first-line treatment, second-line treatment

## Abstract

Clinical outcomes of chemotherapy for patients with advanced pancreatic adenocarcinoma in a real-world setting might differ from outcomes in randomized clinical trials (RCTs). Here we show in a single-institution cohort of 595 patients that median overall survival (OS) of patients who received gemcitabine alone (*n* = 185; 6.6 months (95% CI; 5.5–7.7)) was the same as in pivotal RCTs. Gemcitabine/capecitabine (*n* = 60; 10.6 months (95% CI; 7.8–13.3)) and gemcitabine/nab-paclitaxel (*n* = 66; 9.8 months (95% CI; 7.9–11.8)) resulted in a longer median OS and fluorouracil/oxaliplatin/irinotecan (*n* = 31, 9.9 months (95% CI; 8.1–11.7)) resulted in a shorter median OS than previously reported. Fluorouracil/oxaliplatin (*n* = 35, 5.8 months (95% CI; 4.5–7)) and best supportive care (*n* = 206, 1.8 months (95% CI; 1.5–2.1)) could not be benchmarked against any RCTs. The degree of protocol adherence explained differences between real-world outcomes and the respective RCTs, while exposure to second-line treatments did not.

## 1. Introduction

Pancreatic ductal adenocarcinoma (PDAC) is the fourth leading cause of cancer-related death in the European Union [1]. It is estimated to become the second most deadly cancer by 2030 because of an increasing incidence and a lack of the major improvements in prevention, early detection, and treatment seen in other common malignancies [2].

For the vast majority of patients, who have an unresectable tumor or metastatic disease at diagnosis, and for those who relapse after surgery, cytotoxic chemotherapy remains the best available treatment. Monotherapy with gemcitabine is generally well-tolerated, even by patients with reduced performance status. It alleviates disease symptoms but only marginally improves survival outcomes [3,4]. Combination of gemcitabine with oxaliplatin or capecitabine significantly improves the objective response rate and progression-free survival over gemcitabine alone, but both combinations fail to improve overall survival (OS) in randomized clinical trials (RCTs) [5,6,7]. The combination of gemcitabine with erlotinib is associated with a small, but statistically significant, improvement in OS over gemcitabine alone, but excess adverse events and high costs have limited its use in clinical practice [8]. 

Gemcitabine and nab-paclitaxel show an improved OS of 8.5 months compared to 6.7 months for gemcitabine [9]. The main alternative for patients fit for combination therapy is fluorouracil (5-FU)/oxaliplatin/irinotecan with leucovorine (FOLFIRINOX), which gives a median OS of 11.1 months compared to 6.8 months in a gemcitabine-control arm [10]. In both cases, relatively small improvements of OS come at the price of more toxic regimens, which biases clinical studies towards patients with good performance status. Thus, the majority of patients in an unselected cohort with metastatic PDAC were not eligible for gemcitabine/nab-paclitaxel or 5-FU/oxaliplatin/irinotecan if the respective phase III study’s inclusion and exclusion criteria were applied consistently [11]. However, a recent survey among European physicians involved in the treatment of metastatic pancreatic cancer revealed that 5-FU/oxaliplatin/irinotecan and gemcitabine/nab-paclitaxel are the preferred first-line chemotherapy options in routine clinical practice [12]. 

For patients, whose cancer progresses after first-line treatment, various approaches have been tested, but no consensus has been reached [13]. Best studied is the combination of 5-FU and oxaliplatin after first-line gemcitabine. One RCT showed an improved OS over 5-FU alone [14], while another study reported shorter OS and increased toxicity in the combination group [15]. Liposomal irinotecan is a new second-line option with good tolerability and the combination of 5-FU with liposomal irinotecan has demonstrated improved OS after previous gemcitabine-based therapy [16]. Gemcitabine-based second-line therapy has to our knowledge not been studied in larger RCTs.

We hypothesized that clinical outcomes of patients with pancreatic cancer in routine clinical care differ from those in RCTs. We used a single-institution cohort to evaluate overall survival according to first-line chemotherapy compared to the pivotal RCTs for the respective regimen. To determine which underlying factors are important for differences between RCT and real-world outcomes, we assessed secondary clinical outcomes, protocol adherence, different sequences of first- and second-line treatment regimens, and adverse events.

## 2. Results

### 2.1. Patient Characteristics

We included 595 patients in our analysis and observed a total of 535 deaths during the follow-up period between January 1, 2013, and April 4, 2018. Data from 60 patients were censored with a median follow-up of 404 days (range 61–1506 days). Patient characteristics indicated substantial heterogeneity across groups that received different first-line treatments (Table 1). The proportion of patients with better performance status and younger age was higher among patients who received combination therapies than among patients with gemcitabine or best supportive care (BSC). We also found that most patients who received 5-FU/oxaliplatin had relapsed after previous surgery and adjuvant chemotherapy and that the group treated with 5-FU/oxaliplatin/irinotecan had the highest proportion of patients who underwent exploration. We also observed differences in the metastatic pattern and the distribution of CA19-9 across groups; patients treated with gemcitabine/nab-paclitaxel had the highest median CA19-9 level. Differences in the rate of diabetes and smoking, although statistically significant, did not follow any specific pattern.

Ninety-four patients died while on the initial treatment. After first-line chemotherapy, 148 of the surviving 295 patients received an active second-line regimen while the other 147 patients received BSC. Gemcitabine (*n* = 27), gemcitabine/nab-paclitaxel (*n* = 23), 5-FU/oxaliplatin (*n* = 53), 5-FU/irinotecan (*n* = 19), or 5-FU or capecitabine monotherapy (*n* = 13) were used as second-line treatments (Appendix A). During second-line treatment, 11 patients were censored with a median follow-up of 101 days (range 18–262 days).

### 2.2. Overall Survival According to First-Line Treatment

The median OS of all patients in the cohort was 5.8 months (95% CI, 5–6.5 months). We analyzed how the choice of first-line chemotherapy correlated with overall survival (Figure 1a). Patients who only had BSC had the shortest overall survival (1.8 months, 95% CI; 1.5–2.1). The median OS of patients who received gemcitabine alone (6.6 months, 95% CI; 5.5–7.7) was almost identical to the OS reported for this regimen in several RCTs (Table 2). Gemcitabine/nab-paclitaxel treatment (9.8 months, 95% CI; 7.9–11.8) and gemcitabine/capecitabine (10.6 months, 95% CI; 7.8–13.3) were associated with a longer median OS than previously reported [6,9]. In contrast, patients who received 5-FU/oxaliplatin/irinotecan had a shorter median OS (9.9 months, 95% CI; 8.1–11.7) than previously reported [10].

We stratified patients by the presence or absence of metastases to assess the effect of disease stage on OS. Differences between patients with local vs. metastatic disease were greatest among those who received gemcitabine (8.2 months, 95% CI; 6.6–9.7 vs. 5.7 months, 95% CI; 4.8–6.6), gemcitabine/capecitabine (12.1 months, 95% CI; 8.8–15.4 vs. 8.7 months, 95% CI; 4.6–12.9), or 5-FU/oxaliplatin (8.4 months, 95% CI; 4.0–12.9 vs. 5.2 months, 95% CI; 3.6–6.8), and thus were greatest for regimens where the RCT used for comparison included patients with both disease stages or where, in the case of 5-FU/oxaliplatin, no RCT was available for comparison (Appendix A). Only six patients who received gemcitabine/nab-paclitaxel had locally advanced disease, and inclusion of these patients did not affect the median OS of the whole group. Unexpectedly, among patients treated with 5-FU/oxaliplatin/irinotecan, the median OS was shorter for those with localized disease (9.9 months; 95% CI, 7.0–12.8) compared to those with metastatic disease (10.3 months; 95% CI, 8.6–12.0). While not statistically significant, this difference might have skewed the whole group towards a generally shorter OS—an interesting finding given that survival outcomes for 5-FU/oxaliplatin/irinotecan were worse than previously reported despite inclusion of a group of patients without metastases in our cohort, a characteristic that is generally considered favorable for survival [17].

Monotherapy with gemcitabine is used as the control arm in RCTs of combination regimens in PDAC, and we assessed the potential survival benefits of drug combinations over gemcitabine in our cohort (Figure 1a). When we controlled for the considerable heterogeneity between groups, treatment with gemcitabine/capecitabine (multivariate hazard ratio (HR), 0.57; 95% CI, 0.41–0.80; Table 2), gemcitabine/nab-paclitaxel (multivariate HR, 0.54; 95% CI, 0.38–0.76), or 5-FU/oxaliplatin/irinotecan (multivariate HR, 0.5; 95% CI, 0.31–0.81) were all associated with similar survival benefits over gemcitabine alone. Surprisingly, our observations did not reflect the substantial differences between these three regimens in the underlying RCTs (Table 2). For gemcitabine/capecitabine, this also meant a significantly improved median OS that had not been demonstrated in the RCT by Cunningham et al. [6]. For 5-FU/oxaliplatin, we did not observe any significant survival benefit over gemcitabine. Although this suggests that 5-FU/oxaliplatin might be inferior to other combination therapies, this has not been tested in an RCT. In addition, findings from our cohort might be difficult to generalize because of an overrepresentation of patients who relapsed after surgery in the 5-FU/oxaliplatin group. Our sensitivity analysis indicates, however, that previous surgery did not affect the effect of chemotherapy after relapse (Appendix A and Appendix A).

Because the proportional-hazard assumption appeared violated, we also estimated HRs and 95% CIs derived from flexible parametric survival models (Figure 2). Dynamic modeling of multivariate HRs showed that the survival benefit associated with combination therapies was most pronounced during the first six months after initiation of treatment. For gemcitabine/capecitabine the effect was maintained for more than a year, for gemcitabine/nab-paclitaxel for approximately a year, and for 5-FU/oxaliplatin/irinotecan for about nine months. There was no significant difference in the HR for death between gemcitabine and 5-FU/oxaliplatin. For patients receiving BSC, the HR for death decreased until six months after the start of observation, but almost no difference compared to patients treated with gemcitabine was observed after that time point.

### 2.3. Time to Treatment-Failure, Evaluation at the End of Treatment, and Protocol Adherence

We analyzed the time to treatment failure (TTF), the clinical evaluation at the end of treatment, and protocol adherence to evaluate the implementation of different regimens in our routine-care setting to better understand the differences in OS and survival benefit between our cohort and previously reported RCT.

Patients with advanced PDAC were generally treated with cytotoxic chemotherapy until progress or unacceptable toxicity. With the exception of gemcitabine/nab-paclitaxel (5.1 months; 95% CI, 4.1–6), TTF was approximately three months in all groups (Figure 1b, Table 2). When controlling for covariates, we observed a significantly lower risk to terminate treatment for patients in the gemcitabine/nab-paclitaxel group compared to gemcitabine alone (multivariate HR, 0.62; 95% CI, 0.44–0.87). In contrast, patients who received 5-FU/oxaliplatin had a significantly shorter TTF than the gemcitabine-only group (Table 2). Differences in TTF corresponded with differences in the clinical evaluations at the end of treatment. Patients treated with gemcitabine/capecitabine (53.3%; 95% CI, 40–66.3%) or gemcitabine/nab-paclitaxel (53%; 95% CI, 40.3–65.4%) had the highest proportions of individuals who were treated until disease progression (Table 2). Proportions were lower for 5-FU/oxaliplatin/irinotecan (48.4%; 95% CI, 30.2–66.9%), 5-FU/oxaliplatin (45.7%; 95% CI, 28.8–63.4%), and gemcitabine (36.8%; 95% CI, 29.8–44.1%). In summary, TTF and the risk of treatment failure did not sufficiently explain differences from previously reported progression-free survival and OS in RCTs.

We assessed the degree to which treatment protocols were followed in clinical practice by comparing the number of administered courses of chemotherapy over the treatment period to a hypothetical ideal protocol adherence, i.e., the amount of chemotherapy that would have been given if patients had been treated without any complications or delays (Figure 3). Patients in the gemcitabine/capecitabine (82.1%; R^2^ = 90.9%) and gemcitabine/nab-paclitaxel (79.4%; R^2^ = 76.1%) groups had the highest cumulative adherence to the protocol. Cumulative adherence to gemcitabine (72.4%; R^2^ = 91.1%) was reduced and it was low among patients treated with 5-FU/oxaliplatin/irinotecan (64.9%; R^2^ = 92.2%) and 5-FU/oxaliplatin (58.8%; R^2^ = 8.4%). 

### 2.4. Second-Line treatment

Patients who were eligible for second-line treatment typically received a fluoropyrimidine-based protocol after gemcitabine-based treatment or vice versa. We grouped similar protocols together and explored the possibility that crossing over to an effective second-line treatment affected the observed OS when patients were stratified by first-line therapy. We identified 102 patients who could be clustered according to treatment with gemcitabine, gemcitabine/capecitabine, or gemcitabine/nab-paclitaxel followed by 5-FU/oxaliplatin, 5-FU/irinotecan, 5-FU, or capecitabine (*n* = 75; 24.1% of all patients who had received one of the three first-line regimens) or according to treatment with 5-FU/oxaliplatin/irinotecan or 5-FU/oxaliplatin followed by gemcitabine or gemcitabine/nab-paclitaxel (*n* = 27; 40.9%).

The median OS measured from the end of first-line treatment (Figure 4a) was the same, 5.0 months, for patients with second-line fluoropyrimidine-based (95% CI, 3.8–6.1 months) and gemcitabine-based treatments (95% CI, 2.7–7.2 months). The univariate HR for death among patients who received gemcitabine-based second-line regimens was 0.97 (95% CI, 0.60–1.47), and the multivariate HR was 1.82 (95% CI, 0.89–3.71) after adjusting for sex, age, performance status, BMI, alcohol consumption, smoking, diabetes, surgery, bile duct interventions, tumor stage, tumor grade, and CA19-9 levels. In summary, we did not observe any differences after discontinuation of the first-line treatment.

Over the whole course of treatment, however, we observed a median OS of 12.8 months (95% CI, 10.9–14.8 months) for patients who received a fluoropyrimidine-based second-line protocol (Figure 4b). For patients who received a gemcitabine-based second-line therapy, the median OS was significantly shorter (9.9 months; 95% CI, 10.9–14.8; *p* = 0.008). The univariate HR for death among these patients was 1.91 (95% CI, 1.18–3.11). When we adjusted the HR as above, the multivariate HR was 3.78 (95% CI; 1.78–8.02). 

### 2.5. Adverse Events

Finally, we reviewed the free-text information in patients’ medical records and routine blood work to explore the extent of adverse events (AEs). In total, 298 of 389 patients (76.6%) had AEs of any grade during first-line chemotherapy. Among these patients, we retrospectively identified 426 individual AEs that were categorized into 76 different items according to the Common Terminology Criteria for Adverse Events (CTCAE) 4.03. The majority of AEs were grade 1 (*n* = 94; 22.1%) and grade 2 (*n* = 148; 34.7%); more severe grade 3 AEs (*n* = 123; 28.9%) were still common but life-threatening grade 4 AEs were relatively rare (*n* = 45; 10.6%). We recorded 16 (3.8%) AEs with fatal outcomes. Among the most frequently recorded AEs were hematological AEs, general malaise, gastrointestinal problems, and severe infections. We observed significant differences across treatment groups, but the only identifiable pattern was the restriction of peripheral sensory neuropathy to patients treated with gemcitabine/nab-paclitaxel or 5-FU/oxaliplatin with or without irinotecan (Table 3).

## 3. Discussion

Benchmarking real-world outcomes of cancer treatment against prospective RCTs can help to identify gaps in the delivery of best practice, give insight into underlying clinical challenges, and optimize the use of available treatments [18]. The major finding of our study was that chemotherapy for patients with advanced pancreatic cancer in clinical routine care can achieve survival benefits that are similar to previously published RCTs. However, there were important differences across the five most common treatment regimens and how they relate to the respective RCTs. Interestingly, treatment with gemcitabine in combination with capecitabine resulted in similar survival outcomes as treatment with gemcitabine/nab-paclitaxel and even the more intense 5-FU/oxaliplatin/irinotecan treatment. While this had not been demonstrated for gemcitabine/capecitabine before, these findings are in line with recently published retrospective cohort studies that found no differences in OS or TTF between patients treated with 5-FU/oxaliplatin/irinotecan or gemcitabine/nab-paclitaxel in Korean and North American centers [19,20,21]. All three combination therapies were superior to gemcitabine alone or 5-FU/oxaliplatin as first-line treatments (Figure 1 and Figure 2, Table 2) but their benefit was lost as patients discontinued their treatment and it disappeared around the time most patients were off the respective protocol (Figure 2). A reasonable explanation for these observations is differences in protocol adherence, and we demonstrated that deviations from the outlined treatment regimen are associated with poorer outcomes than previously reported (Figure 4). Similar to the loss of survival benefit after discontinuation of treatment, this finding underscores the value of keeping up active treatment in a real-world setting—a point that has previously been stressed in the adjuvant treatment of PDAC [22]. In summary, our findings discourage intermittent treatment as well as jeopardizing continuous treatment by choosing regimens that patients might not tolerate in the long run.

A higher proportion of patients who had received 5-FU-based first-line regimens crossed over to gemcitabine-based second-line treatment than vice versa, and we observed no significant difference in OS between different protocol groups (Figure 4a). Taken together, we conclude that the differences in OS between different first-line treatments were not attributable to more frequent or more effective second-line treatment. Of note, the OS in our cohort associated with second-line treatment and measured from the discontinuation of first-line treatment was similar to that reported in prospective clinical trials [14,15,16]. The analysis of second-line treatment was limited because we measured OS for second-line therapy from the discontinuation of the previous treatment, because we could not exclude a stricter selection of patients for second-line treatment in clinical routine care than in RCTs, and because we pooled patients who had received similar treatments. Importantly, however, we observed that the sequence of gemcitabine-based therapy after 5-FU/oxaliplatin with or without irinotecan was associated with poorer outcomes than a 5-FU-based treatment after gemcitabine, gemcitabine/capecitabine, or gemcitabine/nab-paclitaxel (Figure 4b).

The aim of this study was to benchmark the survival benefit of chemotherapy for advanced pancreatic adenocarcinoma in a routine care setting, but its observational design has several limitations. The findings might not be generalizable to all clinical settings, and comparisons of nonrandomized groups are inherently vulnerable to confounding. As expected, we observed differences in age and performance status which might reflect confounding by indication. Similarly, higher CA19-9 levels might reflect selection to treatment with gemcitabine/nab-paclitaxel as patients with this poor prognostic marker have been reported to have additional benefit of this combination [23]. In addition, although most data were generated prospectively in the medical records, review and re-categorization of unstructured data might introduce bias. We acknowledge this issue especially for AEs that might have been flawed by the treating physician’s inclination to document events or symptoms relevant to the patient’s quality of life or to the choice of treatment. We also recorded surprisingly few AEs, which might result from underreporting. Thus, we only used descriptive statistics and did not apply any inferential statistics to AEs. 

In summary, we showed that gemcitabine/capecitabine was associated with greater survival benefits than what prospective RCTs have suggested, and it even outperformed gemcitabine/nab-paclitaxel in our cohort. The triple combination of 5-FU/oxaliplatin/irinotecan might remain a good option for fit patients, but protocol adherence is key and suboptimal first-line treatment cannot be salvaged by second-line therapy. For patients unfit for combination therapy, gemcitabine remains the preferred option, while the use of 5-FU/oxaliplatin is discouraged if other options are available. 

## 4. Materials and Methods 

### 4.1. Patient Population

The study population was patients with PDAC or periampullary cancer who presented for an initial visit at one of Karolinska University Hospital’s three sites, at Karolinska Hospital Solna, or Danderyd Hospital between January 1, 2013, and July 31, 2017, and at Södersjukhuset between January 1, 2013, and September 30, 2016. ICD-10 codes C25.x and C24.1 were used to select patients from the electronic record system. Results were matched against the register of patients that had been discussed at our institutional multidisciplinary tumor board to amend the cohort with patients who had not been identified during the initial search. We identified a total of 792 patients. After exclusion of misclassified or incomplete cases, patients who were relapse-free, and patients who had undergone pre-operative chemotherapy or irradiation, we included 595 patients in the analysis (Figure 5). 

### 4.2. Patient and Tumor Characteristics

We recorded general patient and tumor characteristics relevant to clinical outcomes and choice of treatment. All data were retrospectively collected from the electronic medical records. We recorded baseline characteristics (sex, age at diagnosis, diabetes, alcohol abuse, smoking status, BMI, and performance status) at the initial visit. Serum levels of the tumor marker CA 19-9, shown to have prognostic and predictive value in PDAC, measured closest to the initial contact or relapse date were used [24]. Insertion of a biliary stent or drainage, surgical exploration, pancreatic resection, and adjuvant chemotherapy, all of which might have implications for clinical outcomes in an advanced setting, were recorded [25,26]. We retrospectively assessed the clinical stage based on the multidisciplinary tumor board decision, the initial radiology report, and the patient’s referral note according to the American Joint Committee on Cancer (AJCC) Cancer Staging Manual, 7th Edition [27]. For patients who had undergone tumor resection, the pathological staging system (TNM) was used, and clinical information on the M-status was amended at relapse. Tumor morphology and grading were collected from pathology reports or referrals if performed at outside hospitals.

### 4.3. Chemotherapy Protocols

We registered chemotherapy regimens according to our local protocol library. Different protocols for the same drug or drug combination were pooled, and capecitabine could substitute for 5-FU as monotherapy or in doublets with oxaliplatin or irinotecan according to institutional practice. Single drugs or combinations given to fewer than ten patients were summarized as “other”.

### 4.4. Outcome Measures 

For patients with unresectable or metastatic disease at primary diagnosis, OS was measured from the initial physician visit related to the patient’s diagnosis of PDAC until the date of death recorded in the Swedish population register. For patients who had undergone tumor resection, OS was calculated from the date that relapse was documented in the medical record or was brought to our attention if the patient was referred. Patients who were alive and receiving treatment at the hospital or best supportive care (BSC) were censored at the date the record was accessed. Patients who were not reported deceased but with no current information available were considered lost to follow-up and were censored at the time of last contact. TTF was calculated from the initial visit at our department for newly diagnosed unresectable cases or the first return visit for relapse until the visit at which discontinuation of treatment was decided.

### 4.5. Dose Modifications and Protocol Adherence

Any cycle of chemotherapy of which at least one dose was administered was counted. For gemcitabine-based protocols with weekly application for three weeks followed by one week of rest, two partially administered cycles could be pooled into one if they were administered within four weeks. We registered dose-reduction, temporary discontinuation (≤1 cycle) of one drug in combination therapies, and partial application of any cycle except for the last one as protocol modifications. To examine protocol adherence over time, we plotted TTF against the total number of cycles that each patient had received and fitted a linear regression model. For comparison, we also plotted a line that corresponded to an ideal 28-day schedule for gemcitabine-based protocols or an ideal 14-day schedule for 5-FU-based protocols. We then expressed protocol adherence as the relative difference between the slopes of the two lines.

### 4.6. Adverse Events

Adverse events (AE) were assessed per the treating physicians’ notes at follow-up visits and regular blood work. They were initially recorded in a non-standardized fashion, as is routine practice. We used the CTCAE version 4.03 for retrospective classification, and AEs were assigned to the respective treatments during which they occurred [28].

### 4.7. Statistical Analysis

We compared basic characteristics and categorical outcomes across treatment groups with ANOVA, Kruskal–Wallis, and chi-square tests depending on the scale of measurement of the variable. The rates of AEs were compared using the chi-square test. The Kaplan–Meier method and log-rank test were used to analyze OS and TTF. We calculated the HR for death according to treatment protocol with a Cox proportional-hazards model and used multivariate Cox regression to adjust for covariates. We tested the proportional-hazard assumption using the Schoenfeld residuals test. If the assumption appeared to be violated, we estimated HRs and 95% CIs that were derived from flexible parametric survival models, which allowed HRs to change over time. Analyses were performed with SAS v9.4 (SAS Institute, Cary, NC, USA), Stata v14.1 (StataCorp, College Station, TX, USA), and SPSS, v25 (IBM, Armonk, NY, USA). Statistical significance was defined as *p* < 0.05.

### 4.8. Ethics Approval and Consent to Participate

All data handling and analyses involving human subjects were approved by the Ethical Review Board in Stockholm (Etikprövningsnämnden; case no.: 2015/2185-31/4; 2018/986-31/1). We were exempt from obtaining informed consent for this retrospective study of chart data. The register was reported to the hospital’s Data Protection Officer as mandated.

### 4.9. Availability of Data and Material

All data are available from the corresponding author if written approval from the data custodian (Dataskyddsombudet, Karolinska Universitetssjukhuset, Kansliavdelningen, Nya Hemmet, T5, 171 76 Stockholm/Sweden) and the relevant Swedish Ethical Review Board has been obtained. Approvals must be obtained within five years of the establishment of the datasets after which the data will be destroyed according to the originally approved research plan. 

## 5. Conclusions

Chemotherapy for patients with advanced pancreatic cancer in a clinical real-world cohort can achieve survival benefits similar to those described in RCT. The survival benefit associated with gemcitabine/capecitabine was on par with gemcitabine/nab-paclitaxel and greater than previously reported. While evidence from a RCT supports the use of gemcitabine/nab-paclitaxel as the preferred gemcitabine-base first-line combination therapy, our findings warrant reconsideration of gemcitabine/capecitabine for selected patients, e.g., if other combination therapies are no option. 5-FU/oxaliplatin/irinotecan triple therapy remains a good option for well-selected patients, but our findings highlight the importance of good protocol adherence. 5-FU/oxaliplatin did not appear to be an alternative to other combination therapies. Gemcitabine should be chosen in cases where other drugs cannot be used. Our findings do not support that second-line treatment can compensate differences between the outcomes associated with different first-line therapies. 

## Figures and Tables

**Figure 1 cancers-11-01326-f001:**
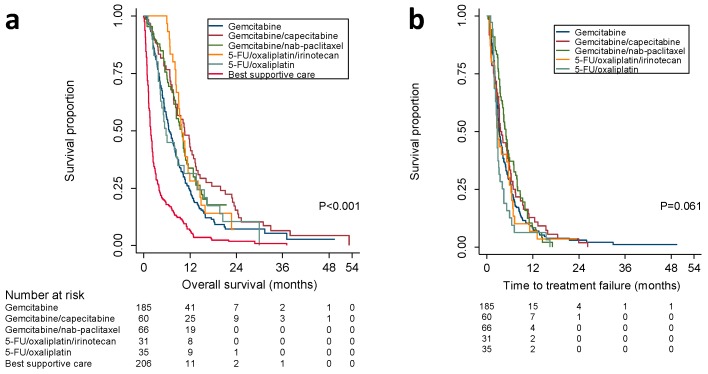
Kaplan–Meier analysis of (**a**) overall survival (OS) and (**b**) time to treatment failure (TTF) according to first-line therapy. OS was calculated from the first visit related to a diagnosis of pancreatic cancer until the date of death. TTF was calculated from the first visit to an oncologist until the visit at which discontinuation was decided. Patients were censored if still alive or under treatment at the date their record was accessed or at last follow-up if no information on the current status was available. Log-rank test; *p* < 0.05 indicates significance.

**Figure 2 cancers-11-01326-f002:**
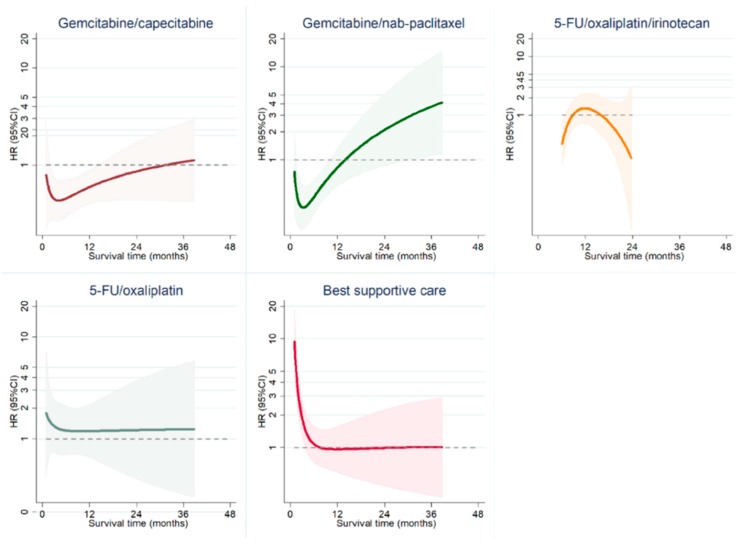
Flexible parametric survival models for HR for death compared to treatment with gemcitabine. All models were adjusted for sex (male or female), age (continuous), BMI (continuous), alcohol consumption (no, current, previously, or unknown), smoking (no, current, previously, or unknown), diabetes (no, yes, or unknown), surgery (no, yes, or unknown), bile duct stenting (no, yes, or unknown), tumor stage (IA/IB/IIA/IIB/III vs. IV), tumor grade (0, 1, 2+), ECOG level (0, 1, 2, 3, or unknown), and CA19-9 level (quantile). Patients taking 5-FU/oxaliplatin/irinotecan were followed up for 24 months because the 95%-CI was large after 24 months. Error bands indicate the 95%-CI.

**Figure 3 cancers-11-01326-f003:**
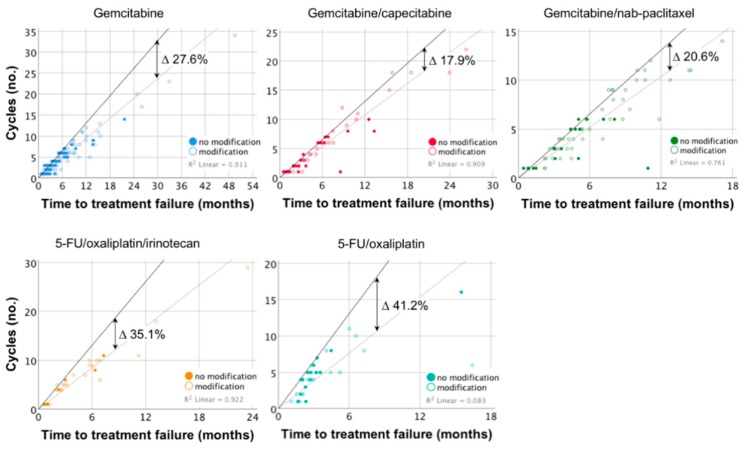
Protocol adherence compared to an ideal treatment schedule according to first-line protocol. The number of administered cycles of chemotherapy was plotted against the TTF for individual patients. Hollow dots indicate protocol modifications. A linear regression model with suppressed intercept was fitted to the plot to illustrate cumulative protocol adherence across each group (dotted line). R^2^ indicates the accuracy of the model. The cumulative adherence for each group was determined in relation to the schedule of the respective protocol (solid line) by comparing the slopes of the lines. If patients terminated treatment or died during an ongoing cycle, their data point could occasionally be plotted to the left of the reference line.

**Figure 4 cancers-11-01326-f004:**
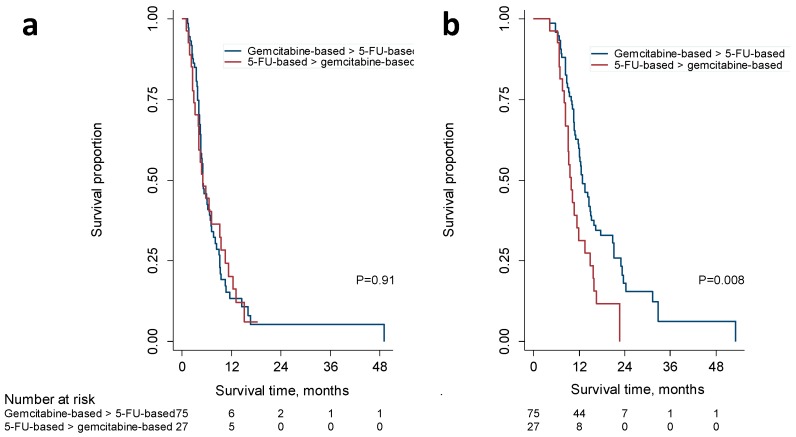
Kaplan–Meier analysis of overall survival associated with second-line systemic chemotherapy (**a**) and the sequence of first- and second-line chemotherapy (**b**). OS was calculated from the discontinuation of first-line treatment (**a**) or the first visit related to a diagnosis of pancreatic cancer (**b**) until the date of death. Patients were stratified into gemcitabine, gemcitabine/capecitabine, or gemcitabine/nab-paclitaxel followed by 5-FU/oxaliplatin, 5-FU/irinotecan, 5-FU, or capecitabine (blue lines) and 5-FU/oxaliplatin/irinotecan or 5-FU/oxaliplatin followed by gemcitabine or gemcitabine/nab-paclitaxel (red lines). Patients were censored if still alive or under treatment at the date their record was accessed or at last follow-up if no information on the current status was available. Log-rank test; *p* < 0.05 indicates significance.

**Figure 5 cancers-11-01326-f005:**
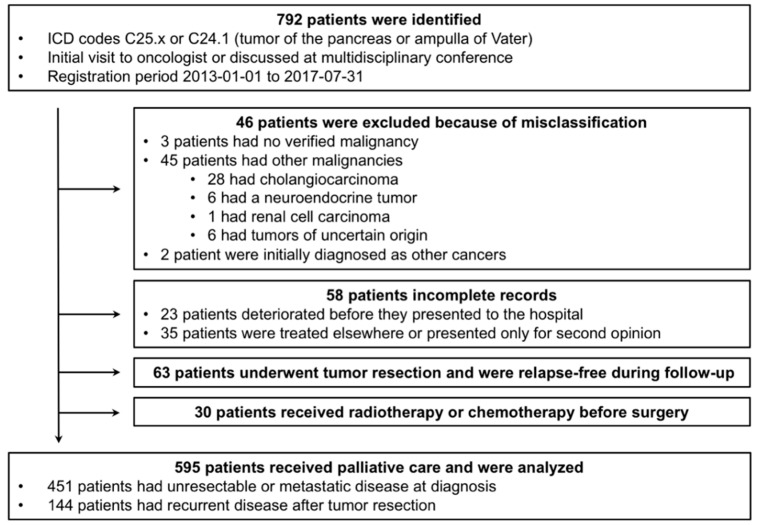
Flow-diagram of patient identification. Patients with a diagnosis of pancreatic cancer or periampullary cancer were selected from the electronic medical record system. Patients who had other diagnoses, who had been misclassified, who had incomplete records, or who had not received treatment for advanced or recurrent pancreatic ductal adenocarcinoma (PDAC) were excluded from the analysis.

**Table 1 cancers-11-01326-t001:** Baseline characteristics of patients according to first-line treatment.

Characteristic	Gemcitabine (*n* = 185)	Gemcitabine/ Capecitabine (*n* = 60)	Gemcitabine/Nab-Paclitaxel (*n* = 66)	5-FU/Oxaliplatin/Irinotecan (*n* = 31)	5-FU/Oxaliplatin (*n* = 35)	Other (*n* = 12)	Best Supportive Care (*n* = 206)	All Patients (*n* = 595)	*p*-Value¶
Sex, no. (%)									
Female	94 (50.8)	25 (41.7)	33 (50)	12 (38.7)	14 (40)	6 (50)	94 (45.6)	278 (46.7)	0.713
Male	91 (49.2)	35 (58.3)	33 (50)	19 (61.3)	21 (60)	6 (50)	112 (54.4)	317 (53.3)	
Age at diagnosis, years									
Mean (range)	70.6 (39.7–83.8)	66.3 (38–81.7)	64.9 (40.5–79.5)	59.4 (39.7–71.9)	65.7 (46.6–76.4)	65.5 (44.2–76.1)	73.1 (51–95.2)	69.4 (38–95.2)	0.000
Body mass index, no. (%) *									
≤18.4	17 (9.2)	6 (10)	3 (4.5)	–	1 (2.9)	–	26 (12.6)	53 (8.9)	0.194
18.5–29.9	157 (84.9)	52 (86.7)	60 (90.9)	30 (96.8)	32 (91.4)	12 (100)	155 (75.2)	498 (83.7)	
≥30.0	10 (5.4)	2 (3.3)	3 (4.5)	1 (3.2)	2 (5.7)	–	15 (7.3)	33 (5.5)	
ECOG PS, no. (%) *									
0	40 (21.6)	26 (43.3)	21 (31.8)	17 (54.8)	12 (34.3)	3 (25)	22 (10.7)	141 (23.7)	0.000
1	88 (47.6)	23 (38.3)	39 (59.1)	12 (38.7)	17 (48.6)	5 (41.7)	49 (23.8)	233 (39.2)	
2	48 (25.9)	7 (11.7)	4 (6.1)	–	3 (8.6)	3 (25)	44 (21.4)	109 (18.3)	
≥3	8 (4.3)	1 (1.7)	–	2 (6.5)	2 (5.7)	1 (8.3)	44 (21.4)	58 (9.7)	
Diabetes, no. (%) * †									
yes	61 (33)	4 (6.7)	17 (25.8)	7 (22.6)	9 (25.7)	2 (16.7)	70 (34)	170 (28.6)	0.002
no	123 (66.5)	56 (93.3)	48 (72.7)	24 (77.4)	26 (74.3)	10 (83.3)	133 (64.6)	420 (70.6)	
Alcohol abuse, no. (%) * †									
yes	10 (5.4)	2 (3.3)	1 (1.5)	–	2 (5.7)	–	18 (8.7)	33 (5.5)	0.414
no	159 (85.9)	53 (88.3)	54 (81.8)	26 (83.9)	33 (94.3)	9 (75)	150 (72.8)	484 (81.3)	
former	6 (3.2)	3 (5)	2 (3)	1 (3.2)	–	–	6 (2.9)	18 (3)	
Smoking, no. (%) * †									
yes	48 (25.9)	12 (20)	9 (13.6)	1 (3.2)	3 (8.6)	–	34 (16.5)	107 (18)	0.003
no	67 (36.2)	24 (40)	23 (34.8)	19 (61.3)	17 (48.6)	4 (33.3)	104 (50.5)	258 (43.4)	
former	61 (33)	21 (35)	27 (40.9)	10 (32.3)	15 (42.9)	5 (41.7)	52 (25.2)	191 (32.1)	
Primary tumor location, no. (%) *									
Head	114 (61.6)	36 (60)	29 (43.9)	15 (48.4)	22 (62.9)	5 (41.7)	100 (48.5)	321 (53.9)	0.046
Body	28 (15.1)	8 (13.3)	14 (21.2)	9 (29)	7 (20)	2 (16.7)	22 (10.7)	90 (15.1)	
Tail	21 (11.4)	4 (6.7)	7 (10.6)	2 (6.5)	3 (8.6)	2 (16.7)	33 (16)	72 (12.1)	
Overlapping	18 (9.7)	8 (13.3)	12 (18.2)	3 (9.7)	–	2 (16.7)	33 (16)	76 (12.8)	
Ampulla of Vater	2 (1.1)	3 (5)	3 (4.5)	2 (6.5)	3 (8.6)	1 (8.3)	10 (4.9)	24 (4)	
Metastasization, no. (%) ‡									
Non-local lymph nodes	12 (6.5)	3 (5)	8 (12.1)	1 (3.2)	1 (2.9)	2 (16.7)	15 (7.3)	42 (7.1)	0.380
Liver	93 (50.3)	27 (45)	37 (56.1)	14 (45.2)	10 (28.6)	6 (50)	119 (57.8)	306 (51.4)	0.047
Lung	31 (16.8)	7 (11.7)	11 (16.7)	1 (3.2)	9 (25.7)	2 (16.7)	29 (14.1)	90 (15.1)	0.258
Peritoneum	20 (10.8)	6 (10)	18 (27.3)	4 (12.9)	7 (20)	4 (33.3)	41 (19.9)	100 (16.8)	0.012
Other	11 (5.9)	–	3 (4.5)	–	5 (14.3)	2 (16.7)	14 (6.8)	35 (5.9)	0.040
No. of metastatic sites, no. (%) ‡									
1	76 (41.1)	30 (50)	44 (66.7)	11 (35.5)	12 (34.3)	5 (41.7)	88 (42.7)	266 (44.7)	0.002
2	23 (12.4)	5 (8.3)	8 (12.1)	2 (6.5)	4 (11.4)	2 (16.7)	34 (16.5)	78 (13.1)	
≥3	7 (4.3)	–	2 (4.5)	1 (3.2)	–	1 (8.3)	8 (4.4)	19 (3.7)	
Morphology, no. (%) *									
Adenocarcinoma	166 (89.7)	50 (83.3)	56 (84.8)	23 (74.2)	33 (94.3)	11 (91.7)	152 (73.8)	491 (82.5)	0.169
Other	–	1 (1.7)	1 (1.5)	1 (3.2)	2 (5.7)	1 (8.3)	5 (2.4)	11 (1.8)	
CA 19–9, kE/l * §									
Median (IQR)	764 (124–5828)	817.5 (76.5–3855.3)	1390 (267–7620)	626 (66–2330)	142 (34.5–881)	170 (79.3–2927.5)	1309.5 (139.3–8975)	908 (106–5814.5)	0.002
Surgery									
Tumor resection, no. (%)	31 (16.8)	5 (8.3)	8 (12.1)	2 (6.5)	30 (85.7)	7 (58.3)	62 (30.1)	145 (24.4)	0.000
Median time to relapse, mo. (IQR)	7.8 (3.9–14.5)	15.7 (3.2–17.3)	4 (2.2–12.9)	14.2 (11.7–16.6)	12.2 (9.8–15.9)	11.5 (9.7–15.6)	8.4 (5.8–12.8)	9.4 (5.8–14.2)	0.069
Adjuvant treatment, no. (%) ‖									
Total	13 (7)	4 (6.7)	5 (7.6)	2 (6.5)	30 (85.7)	7 (58.3)	32 (15.5)	93 (15.6)	0.000
Completed	7 (3.8)	3 (5)	3 (4.5)	2 (6.5)	24 (68.6)	3 (25)	12 (5.8)	54 (9.1)	
Interrupted	6 (3.2)	1 (1.7)	2 (3)	–	6 (17.1)	4 (33.3)	20 (9.7)	39 (6.6)	
Interventions, no. (%)									
ERCP/PTC	84 (45.4)	23 (38.3)	28 (42.4)	20 (64.5)	18 (51.4)	6 (50)	88 (42.7)	267 (44.9)	0.266
Exploration	18 (9.7)	6 (10)	9 (13.6)	7 (22.6)	1 (2.9)	–	6 (2.9)	47 (7.9)	0.000

* Missing cases/percent up to 100 = no information available. † According to physician’s note. ‡ Metastasization at initiation of treatment. Percent missing up to 100 = no distant metastases. § Reference interval <34 kE/l. ‖ Six courses of gemcitabine weekly for three weeks every four weeks. Five patients received adjuvant treatment with gemcitabine/capecitabine. ¶ANOVA: Age; Kruskal–Wallis: CA19-9, time to relapse; chi-square: all other. Single tests were performed for categorical characteristics with mutually exclusive categories, for characteristics with non-mutually exclusive categories individual *p*-values were calculated for each row. Sub-categories were not compared. Abbreviations: ECOG = Eastern Cooperative Oncology Group; ERCP = endoscopic retrograde cholangiopancreatography; IQR = interquartile range; n/a = not applicable; PTC = percutaneous transhepatic cholangiography.

**Table 2 cancers-11-01326-t002:** Overall survival (OS), hazard ratios (HRs) for death, and treatment failure according to first-line regimen.

Outcome	Gemcitabine (*n* = 185)	Gemcitabine/ Capecitabine (*n* = 60)	Gemcitabine/ Nab-Paclitaxel (*n* = 66)	5-FU/Oxaliplatin/ Irinotecan (*n* = 31)	5-FU/Oxaliplatin (*n* = 35)	Other (*n* = 12)	Best Supportive Care (*n* = 206)	*p*-Value
OS								
Median OS, mo. (95% CI)	6.6 (5.5–7.7)	10.6 (7.8–13.3)	9.8 (7.9–11.8)	9.9 (8.1–11.7)	5.8 (4.5–7)	7.9 (2.2–13.7)	1.8 (1.5–2.1)	0.0001
Univariate HR, (95% CI)	1 (ref)	0.67 (0.49–0.91)	0.72 (0.53–0.99)	0.7 (0.46–1.06)	0.95 (0.65–1.41)	–	2.69 (2.18–3.32)	
HR adjusted for co-variables *, (95% CI)	1 (ref)	0.57 (0.41–0.8)	0.54 (0.38–0.76)	0.5 (0.31–0.81)	1.33 (0.84–2.1)	–	2.4 (1.85–3.12)	
Median OS in RCT of first-line treatment, mo. (95% CI)								
Burris et al. [3]	5.7 (–)	–	–	–	–	–	–	
Cunningham et al. [6]	6.2 (5.5–7.2)	7.1 (6.2–7.8)	–	–	–	–	–	
Conroy et al. [10]	6.8 (5.5–7.8)	–	–	11.1 (9–13.1)	–	–	–	
Von Hoff et al. [9]	6.7 (6–7.2)	–	8.5 (7.9–9.5)	–	–	–	–	
HR for death compared to gemcitabine in RCT, (95% CI)								
Cunningham et al. [6]	1 (ref)	0.86 (0.72–1.02)	–	–	–	–	–	
Conroy et al. [10]	1 (ref)	–	–	0.57 (0.45–0.73)	–	–	–	
Von Hoff et al. [9]	1 (ref)	–	0.72 (0.62–0.83)	–	–	–	–	
Time to treatment-failure								
TTF, mo. (95% CI)	3.3 (2.8–3.8)	3.7 (2.4–4.9)	5.1 (4.1–6)	2.9 (2–3.8)	2.8 (2.4–3.1)	3.5 (2.1–4.8)	–	0.08
Univariate HR, (95% CI)	1 (ref)	0.88 (0.66–1.18)	0.78 (0.59–1.05)	1.09 (0.74–1.62)	1.43 (0.99–2.07)	–	–	
HR adjusted for co-variables *, (95% CI)	1 (ref)	0.86 (0.62–1.2)	0.62 (0.44–0.87)	0.95 (0.57–1.57)	1.8 (1.09–2.98)	–	–	
Progression-free survival in RCT of first-line treatment, months (95% CI)								
Burris et al. [3]	3.7 (–)	–	–	–	–	–	–	
Cunningham et al. [6]	3.8 (2.9–4.8)	5.3 (4.5–5.7)	–	–	–	–	–	
Conroy et al. [10]	3.3 (2.2–3.6)	–	–	6.4 (5.5–7.2)	–	–	–	
Von Hoff et al. [9]	3.7 (3.6– 4)	–	5.5 (4.5–5.9)	–	–	–	–	
HR for disease progression compared to gemcitabine in RCT, (95% CI)								
Cunningham et al. [6]	1 (ref)	0.78 (0.66–0.93)	–	–	–	–	–	
Conroy et al. [10]	1 (ref)	–	–	0.47 (0.37–0.59)	–	–	–	
Von Hoff et al. [9]	1 (ref)	–	0.69 (0.58–0.82)	–	–	–	–	
Clinical evaluation at end of treatment								
Progression, no. (%)	68 (36.8)	32 (53.3)	35 (53)	15 (48.4)	16 (45.7)	–	–	0.002
Stable disease, no. (%)	23 (12.4)	4 (6.7)	5 (7.6)	3 (9.7)	2 (5.7)	–	–	
Partial response, no. (%)	7 (3.8)	5 (8.3)	8 (12.1)	2 (6.5)	3 (8.6)	–	–	
Mixed response, no. (%)	1 (0.5)	1 (1.7)	3 (4.5)	2 (6.5)	5 (14.3)	–	–	
Death, no. (%)	54 (29.2)	5 (8.3)	10 (15.2)	0 (0)	7 (20)	–	–	
Not evaluated, no. (%)	32 (17.3)	13 (21.7)	5 (7.6)	9 (29)	5 (14.3)	–	–	

* The multivariate Cox regression models were adjusted for sex (male or female), age (continuous), BMI (continuous), alcohol consumption (no, current, previously, or unknown), smoking (no, current, previously, or unknown), diabetes (no, yes, or unknown), surgery (no, yes, or unknown), bile duct stenting (no, yes, or unknown), tumor stage (IA/IB/IIA/IIB/III vs. IV), tumor grade (0, 1, 2+), ECOG level (0, 1, 2, 3, or unknown), and CA19-9 level (quantile). OS and TFF were compared using the Kaplan–Meier method and log-rank test. Rates of clinical results at the end of treatment were compared using the chi-square test. Clinical outcomes were compared using chi-squared test. *p* < 0.05 was considered statistically significant. “(ref)” indicates that this column is the reference for the statistical tests and thus the hazard that the HRs in the other columns in the same row refer to.

**Table 3 cancers-11-01326-t003:** Adverse events (all grades) accounting for ≥5% of total recorded events.

Adverse event (CTCAE 4.02)	Gemcitabine (*n* = 185)	Gemcitabine/ Capecitabine (*n* = 60)	Gemcitabine/ Nab-Paclitaxel (*n* = 66)	5-FU/ Oxaliplatin/ Irinotecan (*n* = 31)	5-FU/ Oxaliplatin (*n* = 35)	Other (*n* = 12)	All Treated Patients (*n* = 389)	*p*-Value
Hematological adverse events, no. (%)								
Anemia								
All grades	15 (8.1)	3 (5)	17 (25.8)	5 (16.1)	2 (5.7)	–	42 (10.8)	0.000
Grade ≥3	(0)	(0)	3 (4.5)	2 (6.5)	1 (2.9)	–	(0)	
Platelet count decrease								
All grades	20 (10.8)	3 (5)	14 (21.2)	2 (6.5)	1 (2.9)	–	38 (9.8)	0.000
Grade ≥3	10 (5.4)	1 (0.5)	2 (1.1)	(0)	(0)	–	13 (7)	
White blood cell decrease								
All grades	11 (5.9)	3 (5)	14 (21.2)	2 (6.5)	–	–	38 (9.8)	0.004
Grade ≥3	2 (1.1)	2 (1.1)	2 (1.1)	–	–	–	6 (3.2)	
Non-hematological adverse events, no. (%)								
Bile duct obstruction								
All grades	6 (3.2)	4 (6.7)	3 (4.5)	1 (3.2)	–	–	14 (3.6)	0.042
Grade ≥3	6 (3.2)	4 (6.7)	3 (4.5)	1 (3.2)	–	–	14 (3.6)	
Diarrhea								
All grades	6 (3.2)	3 (5)	4 (6.1)	5 (16.1)	3 (8.6)	1 (8.3)	22 (5.7)	0.000
Grade ≥3	2 (1.1)	1 (1.7)	–	2 (6.5)	2 (5.7)	1 (8.3)	8 (2.1)	
Fatigue								
All grades	19 (10.3)	2 (3.3)	2 (3)	2 (6.5)	2 (5.7)	2 (16.7)	29 (7.5)	0.000
Grade ≥3	5 (2.7)	–	–	–	–	–	5 (1.3)	
Fever								
All grades	8 (4.3)	2 (3.3)	5 (7.6)	1 (3.2)	1 (2.9)	–	17 (4.4)	0.04
Grade ≥3	3 (1.6)	–	2 (3)	–	–	–	5 (1.3)	
Nausea								
All grades	8 (4.3)	3 (5)	4 (6.1)	1 (3.2)	1 (2.9)	–	17 (4.4)	0.08
Grade ≥3	4 (2.2)	2 (3.3)	–	–	–	–	6 (1.5)	
Peripheral sensory neuropathy								
All grades	–	–	14 (21.2)	2 (6.5)	1 (2.9)	–	38 (9.8)	0.000
Grade ≥3	–	–	3 (1.6)	–	–	–	3 (1.6)	
Sepsis								
All grades (always Grade ≥4)	15 (8.1)	–	6 (9.1)	2 (6.5)	4 (11.4)	–	27 (6.9)	0.000

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
