# Peer review of "Survival Benefits of Chemotherapy for Patients with Advanced Pancreatic Cancer in A Clinical Real-World Cohort"

_cancers, 2019, doi:10.3390/cancers11091326_

Round 1
Reviewer 1 Report
This is a study of benchmarking real-world outcomes of palliative pancreatic cancer chemotherapy against prospectively already done RCTs. This study seems to be a very important issue for giving insight into optimizing the use of available treatment options despite of comparison of nonrandomized groups in this study.
1.Patients in the gemcitabine / capecitabine and gemcitabine/nab-paclitaxel resulted in significantly longer median OS than previously reported, and both groups had the highest adherence to the protocol. Despite of these results in this study, why did the authors describe that "gemcitabine/capecitabine was associated with a greater survival benefit than previously reported supporting its use"? In this conclusion session, possible usefulness of gemicitabine/nab-paclitaxel should be described similarly with gemcitabine/capecitabine.
2. It was shown in this study that there was no significant difference in OS between different protocol groups including the second-line treatment shown in Figure 4-a. This result mght be able to suggest that second-line treatment could cancel the OS difference induced by the first-line treatment in this study fashion. However , authors concluded that this study findings do not support that second-line treatment can compensate difference between the outcomes associated with different first-line therapies(page 14, line 382-384. I could not understand these interpretations of the results in this study.
Author Response
Point 1: “Patients in the gemcitabine / capecitabine and gemcitabine/nab-paclitaxel resulted in significantly longer median OS than previously reported, and both groups had the highest
adherence to the protocol. Despite of these results in this study, why did the authors describe that "gemcitabine/capecitabine was associated with a greater survival benefit than previously
reported supporting its use"? In this conclusion session, possible usefulness of gemicitabine/nab-paclitaxel should be described similarly with gemcitabine/capecitabine.”
Response 1: We have clarified in the conclusion section that the OS and survival benefits compared to gemcitabine were similar for gemcitabine/capecitabine and gemcitabine/nabpaclitaxel (lines 432-434).
The original conclusion intended to highlight that we observed a significant survival benefit for gemcitabine/capecitabine compared to gemcitabine that had not been observed in the trial by Cunningham et al. – which in our opinion encourages reconsideration in selected clinical cases. We do, however, still consider gemcitabine/nab-paclitaxel the preferred gemcitabinebased combination treatment because its use is supported by stronger evidence from a prospective RCT. We rephrased and amended the relevant sentence for better clarity (lines 434–436).
Point 2: “It was shown in this study that there was no significant difference in OS between different protocol groups including the second-line treatment shown in Figure 4-a. This result mght be able to suggest that second-line treatment could cancel the OS difference induced by the first-line treatment in this study fashion. However , authors concluded that this study findings do not support that second-line treatment can compensate difference between the
outcomes associated with different first-line therapies(page 14, line 382-384. I could not understand these interpretations of the results in this study.”
Response 2: We demonstrate, as shown in figure 4a and stated in lines 242–244, that there was no difference in OS associated with different second-line treatment approaches measured from discontinuation of first-line treatment. The OS was similar independent of previous
treatment and we consequently conclude that second-line treatment cannot ‘salvage’ differences in OS associated with the choice of first-line regimen. We further illustrate this finding in figure 4b, which summarizes the OS associated with the complete treatment sequences and measured from inclusion in our cohort. We reviewed our conclusion that “Our findings do not support that second-line treatment can compensate differences between the outcomes associated with different first-line therapies.” (lines 440–441) but we cannot understand why reviewer 1 does not share our interpretation. Because we consider the conclusion correct we did not make any content changes but added text for additional clarity in the results section (lines 248–249; 251; 263).
Reviewer 2 Report
Generally speaking, this is an interesting study and it should make contributions to this field. The following concerns need to be addressed before it is accepted.
The p value in the tables needs to be articulated clearer. For example, how the authors made the comparisons. For the parameters listed in the tables, the authors need to explain in more details. For example, CA 19-9, the authors need to make comparisons for all groups and explain if it has some implications. The authors can cite some references (Heger et al., HPB (Oxford), 2019; Xing et al., Gastroenterol Res Pract, 2018; Wu et al., Clin Adv Hematol Oncol. 2013) to explain further.
Author Response
Point 1: “The p value in the tables needs to be articulated clearer. For example, how the authors made the comparisons.”
Response 1: We specified more clearly in the methods section how p-values for characteristics, outcomes and AEs were calculated across treatment groups (line 407, 409). We also included this information in table 1and 2 where different tests were used. Comparisons across groups in tables 3 were made using chi-square tests, evident from the method section, and this information was not included in table 3 to keep it simple.
Point 2: “For the parameters listed in the tables, the authors need to explain in more details. For example, CA 19-9, the authors need to make comparisons for all groups and explain if it has some implications. The authors can cite some references (Heger et al., HPB (Oxford), 2019; Xing et al., Gastroenterol Res Pract, 2018; Wu et al., Clin Adv Hematol Oncol. 2013) to explain further.”
Response 2: It is not exactly clear what reviewer 2 means with “comparisons of all groups and […] if it [sic] has some implications.” We understand that reviewer 2 asks for testing of all
characteristics in pairs of two. For table 1 this would require 255 tests (15 possible combinations across six groups multiplied by 17 characteristics). We consider the ensuing multiple testing problematic and disagree with reviewer 2 that such testing will add additional
information since the purpose of table 1 is to outline the study population and to broadly reveal skewed distributions of potential confounders.
One suggested article investigates the role of CA19-9 after pre-operative/neoadjuvant chemotherapy and the two others deal with its role for detection of pancreatic cancer. A detailed investigation of the role of CA 19-9 in our cohort was, however, not the scope of this study.
We have, however, added additional information on the clinical parameters as asked for by reviewer 2 and especially addressed the role of CA19-9 for patients’ prognosis and potential implications of higher CA19-9 levels in some groups (lines 85–89 and 328–332). As part of these changes, we also cited an additional new reference.
Reviewer 3 Report
The authors report on clinical benefits of advanced pancreatic cancer patients from a single institution and compare their results to previously published RCTs. The data is interesting but not surprising- variation in clinical benefit is expected especially when comparing RWE at single institutions with a different patient population to RCTs. Overall, this analysis is of interest to the field and the authors have done a very good job of organizing the results. Overall the paper is well written but could use some syntax/grammatical corrections. Specifically, Please clarify the following point: Please clarify "palliative treatment". Seems that authors are referring to all treatment as "palliative chemotherapy". I don't think this should be the case for first line treatment. Palliative chemotherapy generally refers to symptom management. I would recommend not using this term unless the therapy was used to manage symptoms.
Author Response
Point 1: “Please clarify "palliative treatment". Seems that authors are referring to all treatment as "palliative chemotherapy". I don't think this should be the case for first line treatment. Palliative chemotherapy generally refers to symptom management. I would
recommend not using this term unless the therapy was used to manage symptoms.”
Response 1: We agree with reviewer 3’s criticism of our use of the term “palliative chemotherapy”. We have changed our language accordingly.
Additionally, we made another language revision with multiple minor changes.

Round 2
Reviewer 2 Report
The authors have not addressed my comments yet.
First, I could not find the changes in lines 407, 409. Second, in Table 1, the P value in the far right, is for comparison for all females vs all males or for genders for a specific treatment? It is confused. For the characteristics of the tables, the authors need to give rationales to use them. I gave an example CA 19-9 in my last comment.
Author Response
Response to Reviewer 2 Comments
ROUND 2
Point 1: “First, I could not find the changes in lines 407, 409.”
Response 1: We apologize for giving the wrong lines in our response. We made changes to
section ‘4.7 Statistical analysis.’ and the correct lines in the latest manuscript are 368 and 370.
Text in original submission: We compared treatment groups with ANOVA, Kruskal–Wallis,
and chi-square tests depending on the scale of measurement of the variable.
More specific description after comments: We compared basic characteristics and categorical
outcomes across treatment groups with ANOVA, Kruskal–Wallis, and chi-square tests
depending on the scale of measurement of the variable. The rates of AEs were compared using
the chi-square test.
Point 2: “Second, in Table 1, the P value in the far right, is for comparison for all females vs
all males or for genders for a specific treatment? It is confused.”
Response 2: The p-value in the right column of table 1 is given for the comparison of all
treatment groups/columns (i.e. we tested if there are any significant differences between the
seven groups). We do this by using different statistical tests that can compare three or more
groups as we outline in our method section ‘4.7 Statistical analysis’.
The choice of test depends on the variable that we compare across groups. For categorical
items, we used the chi-square test. For characteristics for which we give medians, we used the
Kruskal-Wallis test (that extends the Mann–Whitney U test beyond two groups) and for
variables for which we give means, we used ANOVA (ANOVA tests whether the means of
two or more groups are equal, i.e. a t-test for more than two means). All tests assess only if any
of the groups is statistically different from the other(s).
As we described in our earlier response we consider this the adequate statistical test and do not
agree with the suggestion to test all treatment groups pairwise for all characteristics. This kind
of testing is prone coincidental significances due to multiple testing and will in our opinion not
provide relevant additional information. It can, in contrast, be misleading.
We find the information we give on p-values in the method section and in the tables’ footers
clear and transparent after the revision that we made following the previous comments.
Point 3: For the characteristics of the tables, the authors need to give rationales to use them. I gave an example CA 19-9 in my last comment.
Response 3: We did not understand that Reviewer 2 previously requested rationales for the
selection of patient and tumor characteristics in our study. Instead, we detailed the significance
of CA 19-9 in the context of our study. However, we now addressed this issue and included
additional information on the rationales for the use of different variables to characterize our
cohort (lines: 322-323, 325-326 and 328-329). We also cited three additional references
(references 24–26) to this end. In summary, the selected characteristics have previously been
implicated to influence clinical outcomes.
ROUND 1
Point 1: “The p value in the tables needs to be articulated clearer. For example, how the authors
made the comparisons.”
Response 1: We specified more clearly in the methods section how p-values for characteristics,
outcomes and AEs were calculated across treatment groups (line 407, 409). We also included
this information in table 1and 2 where different tests were used. Comparisons across groups in
tables 3 were made using chi-square tests, evident from the method section, and this
information was not included in table 3 to keep it simple.
Point 2: “For the parameters listed in the tables, the authors need to explain in more details. For example, CA 19-9, the authors need to make comparisons for all groups and explain if it
has some implications. The authors can cite some references (Heger et al., HPB (Oxford), 2019;Xing et al., Gastroenterol Res Pract, 2018; Wu et al., Clin Adv Hematol Oncol. 2013) to explain
further.”
Response 2: It is not exactly clear what reviewer 2 means with “comparisons of all groups and
[…] if it [sic] has some implications.” We understand that reviewer 2 asks for testing of all
characteristics in pairs of two. For table 1 this would require 255 tests (15 possible
combinations across six groups multiplied by 17 characteristics). We consider the ensuing
multiple testing problematic and disagree with reviewer 2 that such testing will add additional
information since the purpose of table 1 is to outline the study population and to broadly reveal
skewed distributions of potential confounders.
One suggested article investigates the role of CA19-9 after pre-operative/neoadjuvant
chemotherapy and the two others deal with its role for detection of pancreatic cancer. A detailed
investigation of the role of CA 19-9 in our cohort was, however, not the scope of this study.
We have, however, added additional information on the clinical parameters as asked for by
reviewer 2 and especially addressed the role of CA19-9 for patients’ prognosis and potential
implications of higher CA19-9 levels in some groups (lines 85–89 and 328–332). As part of
these changes, we also cited an additional new reference.
Round 3
Reviewer 2 Report
The revision has been improved.
However, the p-value in Table 1 is still confused.
For example, for gender, it is clear now.
But for smoking, there is one p-value. However, for Metastasization – no., there are 5 p-values. The authors need to make them clearer.
Author Response
ROUND 3
Point 1: “However, the p-value in Table 1 is still confused. For example, for gender, it is clear now. But for smoking, there is one p-value. However, for Metastasization – no., there are 5 p-values. The authors need to make them clearer.”
Response 1: We are sorry that the calculation of the p-values in the right column of table 1 still leaves questions unanswered.
In fact, the latest question raised by Reviewer 2 affects only categorical variables and not continuous variables, like age, time or CA 19-9. The calculation of the p-values for categorical characteristics follows a stringent principle. For characteristics with mutually exclusive categories, e.g. gender with male/female, we only calculate one p-value. The same principle applies e.g. to smoking, because each patient can only be either non-smokers, active smokers or former smokers, or the number of metastatic sites (because only one number can apply to any individual patient). The chi-square test of these characteristics compares the distribution of ³2 categories across the treatment groups. For non-mutually exclusive categories, e.g. the presence/absence of metastases in different
organs, this does not apply. One patient can have metastases in different sites – e.g. liver and lung – and the fact that metastases exist in one organ has no relevance for the other organs. As a consequence, we cannot compare the distribution of presence/absence in different organs at the same time. Thus, we calculate p-values for every single row for this type of
category. We hope to make this procedure transparent and clear by inserting the following additional information in the footer of table 1: “Single tests were performed for categorical characteristics with mutually exclusive categories, for characteristics with non-mutually exclusive categories individual p-values were calculated for each row. Sub-categories were
not compared (lines 84–87).” During our latest revision of the manuscript we coincidentally noticed that the p-value for comparison of the time to relapse after surgery had been lost from table 1 and inserted the
missing value again (line 80).